# Micro-CT Marginal and Internal Fit Evaluation of CAD/CAM High-Performance Polymer Onlay Restorations

**DOI:** 10.3390/polym15071715

**Published:** 2023-03-30

**Authors:** Flavia Roxana Toma, Lavinia Cristina Moleriu, Liliana Porojan

**Affiliations:** 1Department of Dental Prostheses Technology (Dental Technology), Center for Advanced Technologies in Dental Prosthodontics, Faculty of Dental Medicine, “Victor Babeș” University of Medicine and Pharmacy Timișoara, Eftimie Murgu Sq. No. 2, 300041 Timisoara, Romania; 2Department of Functional Science, “Victor Babeș” University of Medicine and Pharmacy Timișoara, Eftimie Murgu Sq. No. 2, 300041 Timisoara, Romania

**Keywords:** PAEK, CAD/CAM, marginal and internal fitness, onlay, micro-CT evaluation

## Abstract

(1) Background: The use of high-performance polymers for fixed restorations requires additional studies regarding their adaptability and processing with CAD/CAM technology. This in vitro study aims to assess the marginal and internal fit of PEEK and PEKK materials using microcomputed tomography. (2) Methods: Twenty-four (*n* = 8) MOD onlays made of PEKK (Pekkton ivory), unmodified PEEK (Juvora medical), and modified PEEK (BioHPP) were investigated. A typodont mandibular left first molar was scanned to achieve 24 resin, 3D printed abutment teeth. The onlays were fabricated with a five-axis milling machine, and after cementation of the specimens, the marginal (MG) and internal gaps (IG) were evaluated at twelve points in the mesio-distal section and thirteen points in the bucco-lingual section using microcomputed tomography. For statistical data analysis, Wilcoxon signed-rank/paired Student *t*-Test, Mann–Whitney/unpaired Student *t*-Test, and one-way ANOVA test were applied. (3) Results: Significant differences (*p* < 0.05; α = 0.05) were reported between the MG and IG for each material for all three polymers and also among two materials in terms of the MG and IG (except Juvora-BioHPP). The highest IG values were recorded in angular areas (axio-gingival line angle) in the mesio-distal section for all the polymers. (4) Conclusions: For all the materials, MG < IG. The type of polymer influenced the adaptability; the lowest marginal and internal gap mean values were recorded for BioHPP. The analyzed polymer used for onlays are clinically acceptable in terms of adaptability.

## 1. Introduction

Adaptability is a relevant factor that can influence the quality, fracture resistance, and longevity of prosthetic restorations; it depends on the frequency and size of gaps determined by morphological disproportion between the restoration and the dental hard tissues [1,2]. The marginal gap (MG) is defined as the discrepancy between the edge of the restoration and the preparation line; the internal gap (IG) is the discrepancy between the inner part of the onlay and the dental structure [3,4]. Larger marginal gaps cause wear or dissolution of the cement due to chemical corrosion and physical fatigue, leading to bacterial infiltration, hypersensitivity, secondary caries, and periodontal disease [5,6]. Poor inner fitness induces a greater thickness of the cement layer and lower support and retention of the restoration, increasing the risk of fracture [7,8]. The fit accuracy can be influenced by the precision of the milling machine, the size and wear of the rotary instruments, the geometry of the restoration, the type of the material, the evaluation method, and the number of measuring locations [9,10,11]; it also depends on the characteristics of the cement and the adhesion established with the polymer and how resistant it is to mechanical and thermal stress; self-etch resin cements are the most used cements, ensuring a durable and effective bond, which can be proven using the chewing fatigue test. It was reported that the material compensates for accumulated stress exerted on the margins [12,13]. However, other studies have shown less marginal adaptation; this situation could be induced by various curing methods, thereby determining the shrinkage of the material and affecting the quality of the marginal seal [14,15].

With the introduction of the first CAD/CAM system for clinical applications, the choice of the best material (polymer or ceramic) for obtaining inlay/onlays from prefabricated blocks was intensely debated [16,17,18]. PMMA-based materials and composite blocks are used as an alternative to silicate ceramics [19,20]. Several studies reported that composite onlays and crowns for molars, fabricated using CAD/CAM technology, demonstrated high fracture resistance, being indicated for long-term restorations [21,22]; PMMA-based materials showed poorer mechanical properties than those of composites [23]. However, it should be taken into account that the thermal and elastic expansion qualities of these are different from those of ceramics and dental hard tissues [24]. The low clinical failure rate, aesthetic appearance, and stabilization of the tooth substance mean that ceramic was recommended as the standard material for CAD/CAM inlays/onlays restorations [25,26]. The excellent properties of thermoplastic high-performance polymers offer new perspectives for restorative and prosthetic applications; they demonstrated ultra-high mechanical and chemical performances and can be substitutes for ceramics and metallic alloys for achieving various fixed and removable restorations [27,28]. These materials are stiff, strong, resistant to hydrolysis at variable temperatures, have no pores or free monomers, and can withstand extreme loads.

The PAEK (polyaryletherketone) family includes high-performance polymers such as PEEK (polyetheretherketone), PEKK (polyetherketoneketone), and AKP (aryl ketone polymer).

Due to their particular characteristics, PEKK materials are placed at the top of the polymer pyramid; Pekkton ivory (PEKK) has a semi-crystalline structure, high flexibility, and very good mechanical properties [29]. Regarding PEEK, many studies reported qualities such as high biocompatibility, temperature resistance, polishing potential, bond strength with luting cements, radiolucency, low plaque affinity, persistence in many different aging environments, and compared to those of glass-ceramics, zirconia, or metallic-alloys, the modulus of elasticity (3–4 GPa) is similar to that of human bone, which make is effective at absorbing shocks and functional stress [30,31,32,33]. To improve their mechanical properties, ceramic-reinforced PEEK with 20% ceramic fillers (Bio HPP) have been developed [34,35]. Reinforced agents contribute to increasing resistance to occlusal loads in order to prevent plastic deformation, cracks, or fractures [36,37].

These materials can be processed and milled from prefabricated blocks using dental CAD/CAM technology systems. An image-capture unit collects data from the prosthetic field and converts them into virtual measurements, resulting in a digital impression [38]. The scanning process can be performed directly and intraorally by the clinician, with the data being acquired from the prepared teeth being used to create a virtual model, or indirectly by scanning the conventional impressions or casts in dental laboratories [39,40]. The properties of computer-aided impressions can be influenced by the scanning method and principles, color displays, or use of anti-reflection powder. The continued development of the CAD/CAM system and software updates in general have led to a decrease in marginal disharmonies [41,42]. The use of high-performance polymers to obtain inlays/onlays is not very widespread, but the exceptional properties of the materials also mean that they are recommended for fixed prosthetics; the fit accuracy has been poorly documented; therefore, additional studies are needed.

Usually, two non-invasive methods are used to assess marginal and internal gaps: the replica technique and micro-computed tomography [43]. The replica method is accepted as a less expensive, reliable, and easy technique; however, some disadvantages have been observed, such as difficult identification of the margins, possible errors resulting from cutting the silicone, a limited number of points that can be measured, and the accuracy is affected by the type of impression material and measurement procedure [44,45,46,47]. Micro-CT, a 3D imaging technique, performs the analysis after cementing the specimens without having to cut them; the samples can be rotated and viewed as virtual slices, which allows the measurement of any parameters and their comparison in different phases of the experiment [48]. The accuracy of the adaptation is usually evaluated at different points located on marginal ridges and axial and occlusal walls of the prepared teeth [49].

The purpose of this in vitro study was to analyze and compare the adaptability (marginal and internal fit) of onlays, both individually for an experimental material and between groups of monolithic high-performance polymers. The null hypotheses were: (1) there are no differences according to the reference point between MG and IG for each polymer; (2) the type of material does not influence the adaptability of the restorations (there is no difference in terms of MG or IG among the groups).

## 2. Materials and Methods

### 2.1. Onlay Preparation Design

A typodont (Standard Model AG-3, Frasaco) mandibular left first molar was prepared for the (mesio-occlusal-distal) onlay using a high-speed device and diamond rotary burs (Preparation Set 4180.314 VPE1, Komet Dental, Lemgo, Germany) by a single clinician, as follows: First, the mesio-occlusal-distal box cavity with a depth of 2 mm and isthmus width = 2 mm was prepared, and then, the proximal box was also prepared: mesio-distal width = 1 mm, bucco-oral width = 3 mm, and occlusal-gingival depth = 3 mm; the convergence angle of axial walls was 6°; 1 mm of the occlusal surface was uniformly reduced, and a 1 mm heavy chamfer on support buccal cusps and a 0.5 mm contra bevel on the oral cusps were created. The typodont preparation was multiplied to achieve the printed resin abutment teeth using a 3D printer; the procedure reduces additional errors that may occur using non-additive methods for multiplication, and the chosen material simulates the mechanical properties of the natural tooth. Subsequently, twenty-four MOD onlays were obtained from three different monolithic high-performance polymers (Pekkton ivory—P; Juvora Natural Dental—J; BioHPP—B) (Table 1). The samples were fabricated according to the manufacturer’s protocol and divided in three groups (*n* = 8).

### 2.2. CAD/CAM

The typodont preparation was scanned (Smart Optics Vinyl HR scanner—Smart Optics Sensortechnik, Bochum, Germany) and designed (Exocad Dental CAD software—Align Technology, Darmstadt, Germany) to achieve twenty-four 3D printed resin abutment teeth (Asiga DentaMODEL Asiga, Sydney, Australia) using a 3D printer (Asiga Max 385 UV, Asiga, Sydney, Australia); the marginal fit was programmed to be 0 µm, and the simulated cement spacer was set at 0.03 mm (30 µm) starting at 1 mm above the margin. The onlays were fabricated with a 5-axis milling machine (Imes Icore 350 iPRO+, Imes Icore, Dental & Medical Solutions, Las Vegas, NV, USA) using water-cooled rotary instruments with sizes of 1 and 2.5 mm.

Each group of restorations was milled from a single disk corresponding to each type of material, and eight samples were obtained. Minor adjustments were performed on occlusal or axial surfaces using water-cooled fine diamond burs (Preparation Set 4180.314 VPE1, Komet Dental, Lemgo, Germany) (Figure 1a).

### 2.3. Cementation Technique

All the restorations were cemented to the resin abutment teeth using a dual-cure, self-adhesive resin cement (RelyX U200, 3M Espe, Seefeld, Germany) with the same procedure according to the manufacturer’s instructions. The auto-mixed cement was spread evenly on the inner surface of the onlay, which was then carefully placed and pressed on the master preparation with a standardized force of 10 N for 10 s, and then light cured for 2 s on the buccal and oral sides. Excess cement was removed using a scalpel and light cured again for 20 s each from four directions. Finally, the samples were slightly polished using a low-speed hand-piece, a polishing brush, and paste (Renfert Polish, Renfert, Hilzingen, Germany) (Figure 1b).

### 2.4. Micro-CT Analysis 

The accuracy was evaluated using a high-resolution scanner (Nikon XT V 160, Minato, Tokyo, Japan), and the generated data were investigated (MyVGL software Volume Graphics, Heidelberg, Germany). The restoration was scanned, and image corrections were made to achieve the best contrast; for each sample, a resolution of 4 µm per pixel was set, and then each scanned sample was reconstructed separately with a resolution of 10 µm per voxel. The microcomputed tomography included sagittal (mesio-distal) and coronal (bucco-lingual) sections for each material (Figure 2a–c). 

Discrepancy measurements were obtained at twelve locations for each sagittal (mesio-distal) section and thirteen locations for each coronal (bucco-lingual) section. To assess the thickness of the cement, a total of twenty-five measurements were taken of one preparation, and two hundred measurements were taken of each material. Marginal fit was estimated as the mean of the resulting values at locations a, m, and n (in both sections), and the internal fit was estimated as the mean from locations b to f (in the mesio-distal sections) and j to l (in the bucco-lingual sections) (Figure 3a,b).

### 2.5. Statistical Analysis

In the first part, using descriptive statistics, the central tendency and dispersion parameters were calculated. By applying the Shapiro–Wilk test, it was established that the data were distributed; in some cases, the data were normally distributed (*p* > 0.05), and in others, they were not normally distributed (*p* > 0.05); therefore, both types of tests (parametrical and non-parametrical) were performed: unpaired Student *t*-Test (for normal distributions)/Mann–Whitney (for not normal distributions) were used for two different groups (to compare the marginal and internal fits of the two materials); one-way ANOVA (normal distributions)/Kruskal–Wallis (not-normal distributions) were used for the analysis of several dependent groups (marginal or internal fit among all three materials); two-way ANOVA (normal distributions)/Friedman (not-normal distributions) test were used to compare more than two different time points to obtain the interaction effect among the material and measurement points (Mg and IG), and Tukey post hoc test was used for multiple comparison of the means. For statistical tests, JASP (v.16.2, University of Amsterdam, Holland) and the IBM SPSS Statistics software (IBM, New York, NY, USA) were used. A significance level of α = 0.05 was set.

## 3. Results

For each specimen, the linear space between the dental preparation and the internal surface of the onlay was assessed at the established measurement points. The mean values of the marginal and internal gaps (µm) were obtained at each reference point for the mesio-distal and bucco-lingual sections of each material (Table 2).

For P, the lowest values were reported at points a (43.75 ± 5.99), k (50 ± 11.18), and h (58.75 ± 12.18), and the highest values were reported at points b (195 ± 26.93), c (197.5 ± 22.78), and f (195 ± 22.36); for J, the lowest values were reported at points a (64.38 ± 12.23) and m (63.13 ± 12.10), and the highest values were reported at points b (197.5 ± 20.16), c (190 ± 25.74), and f (196.25 ± 16.54); for B, the lowest values were reported at points l (30 ± 8.66), k (33.13 ± 8.45), m (35 ± 7.91), and a (61.88 ± 10.73), and the highest values were reported at points c (183.13 ± 26.63) and j (173.75 ± 28.70).

It was observed that for all three materials, the lowest values were recorded at points a (P < J = B), k (B < P < J), and m (B < J < P), followed by l (B < P < J), and the highest values were recorded at points c (B < J < P), b (B < P = J), and f (B < P = J), followed by j (P = J = B). The total mean values of the evaluation points for all materials are shown in Figure 4.

Regarding MG, the lowest value reported for P was point a (43.75 ± 5.99), the lowest value reported for J was point a = m (63.13 ± 12.10), the lowest value reported for B was point m (35 ± 7.91), and the highest values were at point n for all the materials; the lowest IG values were recorded at point k for all the materials, and the highest values were recorded at points c for P (197.5 ± 22.78) and B (183.13 ± 26.63) and b for J (197.5 ± 20.16). The total mean values (µm) for MG and IG for all three polymers are displayed in Table 3 and represented graphically in Figure 5. 

The two-way ANOVA/Friedman test (α = 0.05) reported significant differences (*p* < 0.001) for all three materials when the statistical analysis was performed to compare the values recorded at the reference points (MG and IG) for each material. Following multiple pairwise comparison of the averages (post hoc test), there were significant differences in terms of MG (points a–m–n), except for J at points a–m, and for IG (from points b to j), the data are ordered in the sequence B > P > J, with the note that points b, c, and f present the most significant differences in relation to the other internal points (l, k, g, h, and i) for all the experimental polymers. There was a significant discrepancy between MG and IG in each group (a more evident one for J) (Figure 6a–c). 

When the statistical tests were applied (Table 4) to compare the MG and IG among two (Figure 7a–c) or three materials (Figure 8a,b), it revealed significant differences (*p* < 0.05) between P–J, P–B, and J–B (except MG, *p* > 0.05). The MG and IG for B were significantly smaller than those of J and P, and the IG of J was significantly larger than that of B and P.

## 4. Discussion 

The aim of this study was to evaluate the marginal and internal fits of MOD onlays fabricated by the milling of three different high-performance polymers using CAD/CAM technology and visualized with 3D radiographic micro-CT; the assessment of adaptability was carried out by measuring the thickness of the cement layer at different locations (marginal and internal points) and statistical analysis of the mean values determined in the two (sagittal and coronal) sections both for the polymer and among the materials to present a complete picture of the restoration procedure for master preparation. To obtain clinically relevant information, Groten [50] recommends a number of 50 measurement points, but for a more extensive analysis, we exceeded this number. 

In the current study, significant discrepancies between MG and IG for each experimental material were reported; the results showed that the internal gaps are significantly greater than the marginal gaps are for each material (more evident with Juvora), and additionally, after multiple pairwise comparisons of the average values measured at the reference points, there were significant differences between the marginal gap points (a, m, and n), except for Juvora (a–m with almost equal values) and among the internal gap points (from b to j); the values at points b, c, and f are significant greater than those at points l, k, g, h, and i for all materials; therefore, the first null hypothesis is rejected. A potential explanation for this aspect may be the largest gaps at locations b and c due to the preparations in these areas being geometrically complex with angles, and these were not adapted during fitting. This situation can be attributed to the reduced scanning accuracy, resulting in rounded edges [51,52], or to the milling processing [53]; the diameter of the smallest rotary instruments was 1 mm, which in narrow areas, such as the proximal cavity, can lead to unwanted removal of the material. It is also possible that the accuracy of resin abutment is adversely affected with deformed or sloping surfaces, causing larger internal gaps. It seems that a consequence of this state is the reduction of adaptability and the increase in the size of the cement layer in other locations such as f and j, where higher values were recorded for all the polymers. The measuring points are located on relatively flat surfaces, allowing the good flow and placement of the cement, and the layer should be minimal if the onlay is optimally adapted to the preparation. Lower values were recorded on the axial walls of the preparations at locations h and l for Pekkton, l for BioHPP, and k for all the materials-as a result of the hydraulic pressure applied on the convergent axial walls, which forced the cement towards the occlusal or marginal areas, allowing flow and evacuation until the minimum potential thickness was obtained [54].

The results support the opinion that the type of material, the manufacturing technology, and the cementing technique can influence the fit of the restoration. Perfect adaptability reduces percolation and cement dissolution and helps to maintain gingival health [55,56,57]. CAD/CAM restorations can be affected by the scanning systems; great adaptability depends on the accuracy of scanning [58,59] and the fabricating technique and the milling equipment (size and wear of the rotary instruments) [60,61]; however, no statistical differences were reported when the discrepancies between three samples obtained with a five-axis milling machine were analyzed [62]. Every innovative stage of development in the CAD/CAM system, from its design to mechanical processing, had a positive impact on the marginal and internal fit of the restorations [63,64].

Among the materials, in terms of the MG and IG, significance differences (*p* < 0.05) were reported (except for MG: J–B); BioHPP showed significantly better marginal and internal accuracy than the other two groups did; Pekkton had the largest marginal gaps, and Juvora had the largest internal gaps. J and B belong to the PEEK group—J is a 100% polyetheretherketone and B is a ceramic-reinforced PEEK with 20% ceramic fillers, which improve its mechanical properties; P is a PEKK (polyetherketoneketone) and presents excellent mechanical properties. The compression/flexural strength values for Pekkton = 246 MPa, for filled PEEK > 150 MPa, and for non-filled PEEK = 111 MPa [29]. In this study, it was observed that BioHPP showed the best adaptability, better machinability, and less susceptibility to fracture; it seems that of the three polymers, it can be more easily processed to obtain inlays/onlays from prefabricated blocks. Juvora had a lower flexural strength and hardness than the other two polymers did, and a greater quantity of material was removed from intaglio surface by the rotary instruments during milling. In the current study, it was revealed that the type of material influences the quality of adaptability; therefore, the second null hypothesis is rejected.

The preparation and configuration of the finish line are essential factors that can influence the marginal discrepancy; heavy chamfer preparation for support buccal (functional) cusps and a contra bevel margin for the lingual (non-functional) cusp were used according to investigations that assessed the impact of various marginal lines [65,66]; in other studies, insignificance differences between the chamfer and shoulder preparation were found [67,68]. No significant differences were found when we were comparing the points a–m for Juvora and MG between Juvora and BioHPP. PEEK materials exhibit good marginal machinability, regardless of finish line design; however, at the point n (contra bevel), the highest values among the marginal points were recorded for all the polymers.

Several studies have assessed that a marginal gap less of than 120 µm and internal gap ranging from 20 to 200 µm are suitable for the optimal fit of fixed restorations [69,70]. In the present study, the mean values for cement thickness range from 30 to 200 µm for onlays, which is considered to be clinically acceptable.

The limitation of this study could be that only a few materials were investigated with a single scanner, and it is possible that other systems with different image capture methods and light sources may lead to other results; therefore, further studies are needed to evaluate the adaptability of fixed restorations obtained using CAD/CAM technology. 

## 5. Conclusions

For all the tested specimens related to the investigated reference locations, various levels of adaption were recorded (less marginal ones for Juvora), and the IG values were significantly greater than the MG values were for all three materials.

The type (composition, mechanical properties, and machinability) of polymer influenced the accuracy. The best behavior is related to the ceramic-reinforced PEEK material, followed by unmodified PEKK and PEKK; all the polymers are clinically acceptable in terms of adaptability.

A poor fit in the angular areas affects adaptability in others areas of the restoration differently.

## Figures and Tables

**Figure 1 polymers-15-01715-f001:**
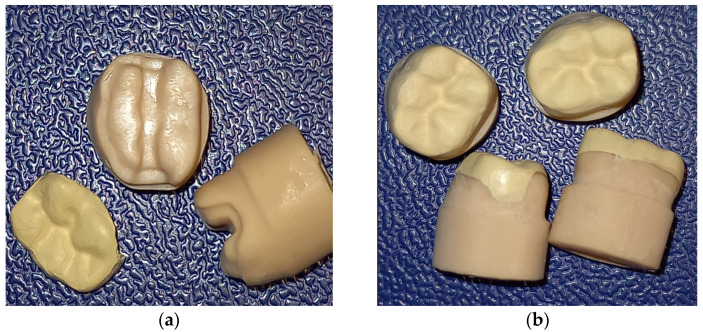
Three-dimensional printed abutments (**a**) and milled polymer onlays (**b**) with cemented onlays.

**Figure 2 polymers-15-01715-f002:**
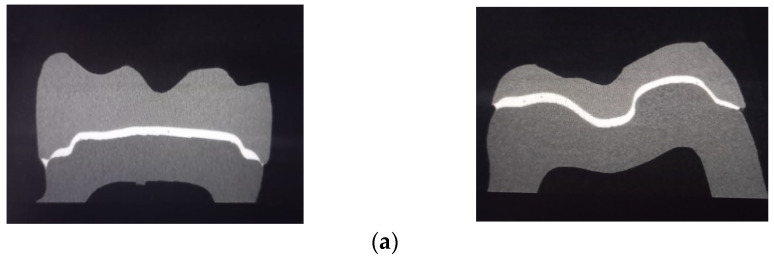
Micro-CT images with sagittal and coronal sections for: (**a**) Pekkton; (**b**) Juvora; (**c**) BioHPP.

**Figure 3 polymers-15-01715-f003:**
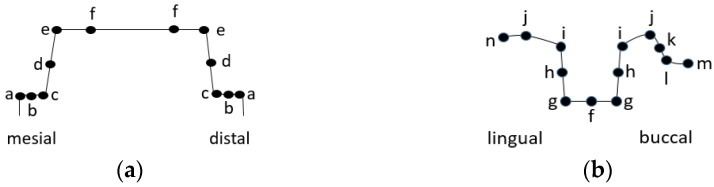
Schematic presentation of measurement location (**a**) in mesio-distal section and (**b**) bucco- lingual section; marginal points = a, m, and n; internal points = b, c, d, e, f, g, h, i, j, k, and l.

**Figure 4 polymers-15-01715-f004:**
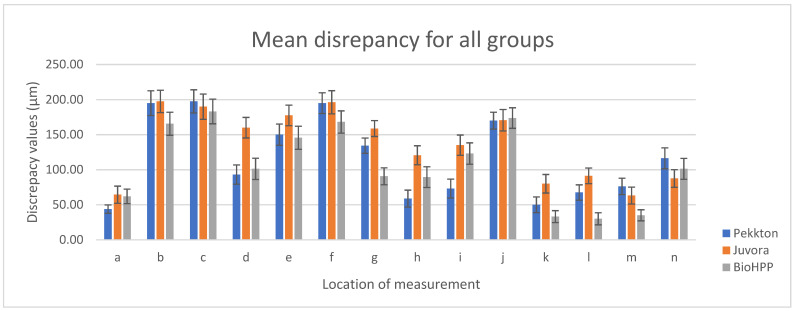
Average values obtained at the reference points of the both sections.

**Figure 5 polymers-15-01715-f005:**
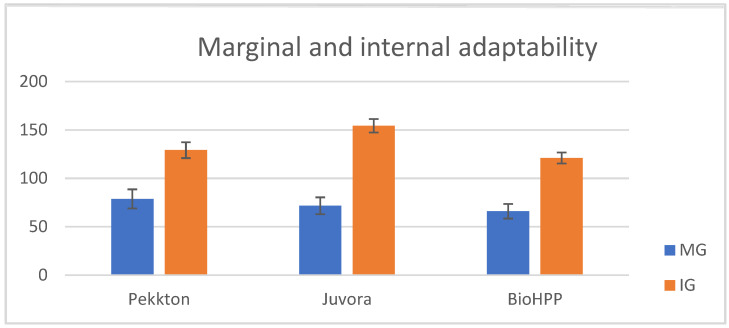
The marginal and internal discrepancy mean values for all three materials.

**Figure 6 polymers-15-01715-f006:**
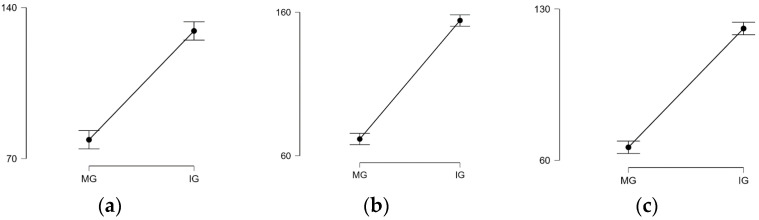
Descriptive plots MG—IG. (**a**) Pekkton; (**b**) Juvora; (**c**) BioHPP.

**Figure 7 polymers-15-01715-f007:**
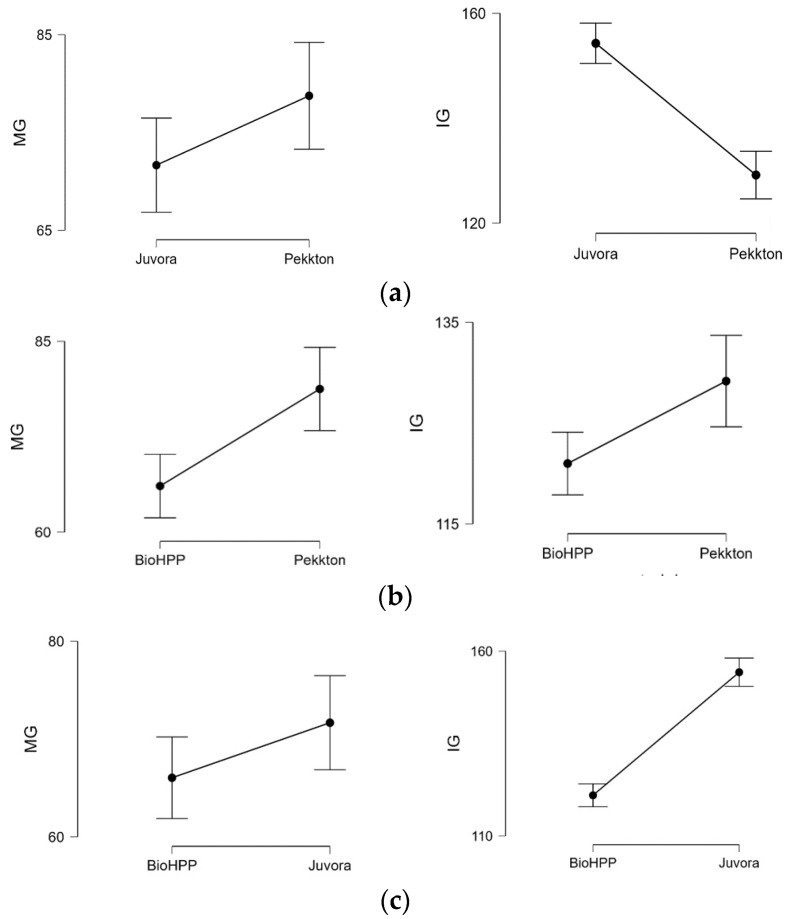
Descriptive plots for MG and MG between materials: (**a**) Juvora-Pekkton; (**b**) BioHPP- Pekkton; (**c**) BioHPP-Juvora.

**Figure 8 polymers-15-01715-f008:**
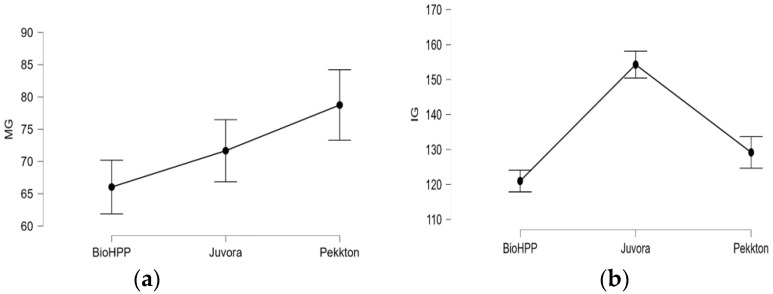
Descriptive plots of (**a**) MG and (**b**) IG for all materials.

**Table 1 polymers-15-01715-t001:** The PAEK materials selected for study (*n* = 8).

Material	Manufacturer	Composition
**Pekkton ivory (P)**	Cendrez + Metaux, Biel/Bienne, Switzerland	polyetherketoneketone (PEKK)
**Juvora Natural Dental (J)**	Juvora Ltd. Global Technology Center, Lancashire, UK	100% polyetheretherketone (PEEK)
**BioHPP shade 2 (B)**	Bredent group GmbH & Co.KG, Senden, Germany	20% ceramic reinforced (0.3–0.5 µm grain size) partially crystalline polyetheretherketone (PEEK)

**Table 2 polymers-15-01715-t002:** Average of linear measurements with standard deviation (SD) at reference points (a↔n); CI = confidence interval (lower–upper bound) (*n* = 8).

	Pekkton (P)		Juvora (J)		BioHPP (B)	
Point	Mean ± SD	95% CI	Mean ± SD	95% CI	Mean ± SD	95% CI
a	43.75 ± 5.99	31.77–55.73	64.38 ± 12.23	39.92–88.84	61.88 ± 10.73	40.42–83.34
b	195 ± 26.93	141.14–248.86	197.5 ± 20.16	157.18–237.81	165 ± 25.73	113.54–216.46
c	197.5 ± 22.78	151.95–243.06	190 ± 25.74	138.52–241.48	183.13 ± 26.63	129.67–236.39
d	93.12 ± 19.60	53.92–132.32	160 ± 20.62	118.76–201.24	101.25 ± 23.15	54.3–147.55
e	150 ± 28.06	93.88–206.12	177.5 ± 24.11	129.28–225.72	145.63 ± 28.50	88.63–202.63
f	195 ± 22.36	150.28–239.72	196.25 ± 16.54	163.17–229.08	168.13 ± 26.98	114.17–222.09
g	134.38 ± 24.99	84.01–83.98	158.75 ± 23.95	110–206.65	90.63 ± 16	58.63–122.63
h	58.75 ± 12.18	34.39–83.11	120.63 ± 21.06	78.51–162.75	89.38 ± 14.78	59.82–118.94
i	73.13 ± 13.56	46.01–100.25	135 ± 22.64	89.72–180.28	123.13 ± 25.67	71.79–174.47
j	170 ± 17.68	134.64–205.36	170 ± 26.09	117.82–222.18	173.75 ± 28.70	116.35–231.15
k	50 ± 11.18	27.64–72.36	80 ± 13.23	53.54–106.46	33.13 ± 8.45	16.23–65.59
l	67.50 ± 10.90	45.7–89.3	91.25 ± 11.11	69.03–113.47	30 ± 8.66	12.68–47.32
m	76.25 ± 11.66	52.93–59.57	63.13 ± 12.10	34.93–87.33	35 ± 7.91	19.18–45.82
n	116.25 ± 22.88	70.49–162.01	87.5 ± 16.39	54.72–120.28	101.25 ± 19	63.25–139.25

**Table 3 polymers-15-01715-t003:** Average values with standard deviation for MG and IG (*n* = 8).

Mean	Pekkton (P)	Juvora (J)	BioHPP (B)
MG	78.75 ± 9.92	71.67 ± 8.74	66.04 ± 7.57
IG	129.17 ± 8.23	154.27 ± 6.98	120.99 ± 9.65

**Table 4 polymers-15-01715-t004:** *p* values regarding MG and IG among polymer groups (*n* = 8).

	MG	IG
P-J	*p* = 0.047	*p* < 0.001
P-B	*p* < 0.001	*p* = 0.013
J-B	*p* = 0.11	*p* < 0.001

## Data Availability

Not applicable.

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
