# Peer review of "Micro-CT Marginal and Internal Fit Evaluation of CAD/CAM High-Performance Polymer Onlay Restorations"

_polymers, 2023, doi:10.3390/polym15071715_

Round 1

Reviewer 1 Report

Interesting study. Please take into account the following advices:

-Please watch out that an aparent note from the editorial team (or a typo from authors) is visible as part of the text at the end of the introduction (lines 94-102).

-In the statistical analysis it is stated that the location of the gap was considered as a variable, and authors compared within them, besides material type. However, this is not reflected neither in the purpose of the work, as it says that the purpose was to "compare the adaptability (marginal and 90 internal fit) of three monolithic high performance polymers" only, and authors are not stating that the location is a study factor too.

-Related to the last comment, the statistical analysis then is not clear. First if non-parametric and parametric statistical tests were employed, authors should provide which part of the data were parametric (with the respectives p values of normality and homoscedasticity tests) to inform the reader which test was applied to which data. However, the most appropriate statistical approach for this study design is a two-way ANOVA to the whole data set together (material type vs. location), of course applying a previous confirmation of normality and homoscedasticity, and a Tukey post-hoc test afterwards to compare between groups, and if data resulted non-parametric, then a kruskall-Wallis to each factor sparated. So, please correct this statistical approach.

Author Response

Review 1

Thank you very much for the review!

-Please watch out that an aparent note from the editorial team (or a typo from authors) is visible as part of the text at the end of the introduction (lines 94-102).

We agree with this point and we removed the inappropriate paragraph.

-In the statistical analysis it is stated that the location of the gap was considered as a variable, and authors compared within them, besides material type. However, this is not reflected neither in the purpose of the work, as it says that the purpose was to "compare the adaptability (marginal and 90 internal fit) of three monolithic high performance polymers" only, and authors are not stating that the location is a study factor too.

We agree with this point and we added additional information:

“The purpose of this in vitro study was to analyse and compare the adaptability (marginal and internal fit) of onlays, both individually for an experimental material and between groups…. The null hypotheses were: 1) there is no differences according to the reference point between MG and IG for each polymer...”

-Related to the last comment, the statistical analysis then is not clear. First if non-parametric and parametric statistical tests were employed, authors should provide which part of the data were parametric (with the respectives p values of normality and homoscedasticity tests) to inform the reader which test was applied to which data. However, the most appropriate statistical approach for this study design is a two-way ANOVA to the whole data set together (material type vs. location), of course applying a previous confirmation of normality and homoscedasticity, and a Tukey post-hoc test afterwards to compare between groups, and if data resulted non-parametric, then a kruskall-Wallis to each factor sparated. So, please correct this statistical approach.

We agree with this point and we added explanations regarding the statistical tests

“Applying the Shapiro-Wilk test, it was established that the data are mixt distributed, in some cases the data were normally (p>0.05), and in others they were not normally distributed (p > 0.05); therefore, both types of tests (parametrical and non-parametrical) were performed: ..“

Reviewer 2 Report

The Manuscript by Flavia Roxana Toma, Lavinia Cristina Moleriu and Liliana Porojan provides of how the onlays were fabricated using a 5-axis milling machine and cemented accordingly. The marginal and internal gaps of such onlays were then evaluated using microcomputed tomography. To determine the accuracy of differences between the regions, a statistical data analysis was deployed. In my considered opinion, the significance of this paper is high as it may find applications in the manufacture of dental prostheses. However, the manuscript needs more work in organisation and details as outlined below.

 Major points:

1)    The authors need to re-write the introduction by highlighting the current study of marginal and internal gaps of different dental materials, and end with the limitations or gaps. They should then demonstrate how their study fits with the identified problem and how they plan to address them.

2)    The authors need to re-write the materials and method section by describing the experimental protocol in detail referring to material and instrument suppliers in parentheses, when possible, to achieve the stated objective without including the results or discussions in this section.

3)    The authors need to re-write the discussions section as follows: First, restate the overall purpose of the study. Then explain the main finding as related to the overall purpose of the study. Next, summarize any interesting findings from the results section. Explain how the statistical findings relate to that purpose of the study. Also describe how the observed results are related to other studies in the field in general. All explanations must be supported by the results of the data analysis.

4)    The authors are encouraged to re-write the conclusion by briefly summarizing the overall conclusions of the study based on the outline of their stated aims and objectives without using pullet points. They should end with the importance of the major finding and its implication to the field of research.

Minor points:

5)    For lines 94 - 102. I would suggest that these sentences are deleted as this highlights the guidance of writing the manuscript.

6)    For lines 106 - 113. I would suggest that the authors should provide a clear procedure for making onlays using a high-speed device and diamond burrs without using pullet points in their descriptions.

7)    For Figure 1a, b and Figure 2a, b & c. I would suggest that these figures should be included in in parentheses or described appropriately for clarity.

8)    For all Figures. I would suggest that the authors should use one format to maintain consistency in labelling.

9)    For the materials and methods. I would suggest that the authors should consider including the resolution of the measurements to justify the two-decimal points accuracy in the result tables.

10) For all the result tables. I would suggest that the authors should include the number of measurements per group for the standard deviation to make sense.

11) For the results section. I would suggest that the authors should provide the justification of why it is necessary to use three different statistical tests.

12) For lines 265 – 267. I would suggest that these sentences should be moved to the materials and methods section as they highlight the process of making typodont.

13) For lines 265 – 267. I would suggest that these sentences should be addressed after discussing their own results as a comparison to demonstrate the accuracy of their findings.

14) Finally, a consistent margin should be applicable for the figures, tables, and text.

Author Response

 Review 1

Thank you very much for the review!

Major points:

  • The authors need to re-write the introduction by highlighting the current study of marginal and internal gaps of different dental materials, and end with the limitations or gaps. They should then demonstrate how their study fits with the identified problem and how they plan to address them.

We agree with this point and we added additional information: “…the choice of the best material (polymer or ceramic) for obtaining inlay/onlays from prefabricated blocks was intensely debated ….. PMMA-based materials and composite blocks are used as an alternative to silicate ceramics … composite onlays and crowns for molars, fabricated using CAD/CAM technology, demonstrated high fracture resistance, being indicated for long-term restorations… the PMMA-based materials showed lower mechanical properties than those of the composites...…recommended ceramic as the standard material for CAD/CAM inlays/onlays restorations

“The excellent properties of thermoplastic high-performance polymers offer new perspectives for restorative and prosthetic applications… they demonstrated ultra-high mechanical and chemical performances and can substitute ceramics and metallic alloys…. “

  • The authors need to re-write the materials and method section by describing the experimental protocol in detail referring to material and instrument suppliers in parentheses, when possible, to achieve the stated objective without including the results or discussions in this section.

We agree with this point and we wrote the suppliers of msterisls and tools in parentheses and removed the   information that belonged to other section..

  • The authors need to re-write the discussions section as follows: First, restate the overall purpose of the study. Then explain the main finding as related to the overall purpose of the study. Next, summarize any interesting findings from the results section. Explain how the statistical findings relate to that purpose of the study. Also describe how the observed results are related to other studies in the field in general. All explanations must be supported by the results of the data analysis.

We agree with this point and we tried to re-write the discussions section as suggested..

  • The authors are encouraged to re-write the conclusion by briefly summarizing the overall conclusions of the study based on the outline of their stated aims and objectives without using pullet points. They should end with the importance of the major finding and its implication to the field of research.

We agree with this point and we to re-write the conclusion section as suggested..

Minor points:

  • For lines 94 - 102. I would suggest that these sentences are deleted as this highlights the guidance of writing the manuscript.

We agree with this point and we deleted this paragraph

  • For lines 106 - 113. I would suggest that the authors should provide a clear procedure for making onlays using a high-speed device and diamond burrs without using pullet points in their descriptions.

We agree with this point and we removed the pullet points from the descriptions: “…by a single clinician, as follows: first, the mesial-occlusal-distal box cavity with depth of 2 mm and isthmus width= 2 mm, then, the proximal box was symmetrically prepared with mesial-distal width=1mm, buccal-oral width=3mm and occlusal-gingival depth=3mm… “

  • For Figure 1a, b and Figure 2a, b & c. I would suggest that these figures should be included in in parentheses or described appropriately for clarity.

We agree with this point and we have included the Figure in parentheses

  • For all Figures. I would suggest that the authors should use one format to maintain consistency in labelling.

We agree with this point and we have included the Figure in parentheses

  • For the materials and methods. I would suggest that the authors should consider including the resolution of the measurements to justify the two-decimal points accuracy in the result tables.

We agree with this point and we corrected the inormation “…for each sample a resolution of 4 µm per pixel was set and then each scanned sample was reconstructed separately with a resolution of 10 µm per voxel…“

  • For all the result tables. I would suggest that the authors should include the number of measurements per group for the standard deviation to make sense.

We agree with this point and we included the number of measurements per group for all the result tables: “ (n=8) “.

  • For the results section. I would suggest that the authors should provide the justification of why it is necessary to use three different statistical tests.

      We agree with this point and we added explanation about the statistical tests

“Applying the Shapiro-Wilk test, it was established that the data are mixt distributed, in some cases the data were normally (p>0.05), and in others they were not normally distributed (p > 0.05); therefore, both types of tests (parametrical and non-parametrical) were performed: “

  • For lines 265 – 267. I would suggest that these sentences should be moved to the materials and methods section as they highlight the process of making typodont.
  • For lines 265 – 267. I would suggest that these sentences should be addressed after discussing their own results as a comparison to demonstrate the accuracy of their findings.

We agree with this point and we moved this paragraph to the materials and methods section

  • Finally, a consistent margin should be applicable for the figures, tables, and text.

      We agree with this point and we made the suggested corrections

Reviewer 3 Report

Dear author , the study looks to be interesting and appreciate the efforts of the authors. But the article needs significant correction in means of grammatical and continuity of the sentences /paragraphs.

Author Response

Review 3

Thank you very much for the review!

Introduction:

 Please go through the introduction and modify it accordingly.

 For e.g.:

 Line 41 – 42 , justification of the sentence needs a citation.

We agree with this point and we decided to remove this paragraph

 Line 43 , the sentence many studies, needs modification ( ass more number of citations) as the sentence starts as many studies

We agree with this point and we corrected the information and moved it to the discussion section

 Line 76 : Polimers : Polymers

We agree with this point and we corrected the information: “…use of high performance polymers to…“

 Parapgraph from 94 to 102, does not imply to your study

We agree with this point and we removed this paragraph

 Methods:

 Lot of spelling errors in materials and methods. [ I feel the methodology needs extensive editing, grammatical as well as clarity in the methodology]

For ex: line 106, [burrs]

 Line 111 [ delete “with”]

Line 118 [ divided ]

 Line 130 : 2,5

Line 140 [ carefully ]

We agree with this points and we corrected the spelling erors

 Discussion:

 The discussion is poorly written. Compare the results of the present study with the previous studies. Maintain the continuity of the paragraph.

We agree with this point and we re-wrote a large part of the discussion section

Round 2

Reviewer 2 Report

After carefully reading the revised version and the rebuttal of the manuscript "Micro-CT Marginal and Internal Fit Evaluation of CAD/CAM High Performance Polymers Onlay-Restorations" by Flavia Roxana Toma, Lavinia Cristina Moleriu and Liliana Porojan, I have decided to recommend for its publication.

Reviewer 3 Report

The authors have addressed the comments.